# Multi-view Framework for Histomorphologic Classification

**Author(s) names withheld**                                    EMAIL(S) WITHHELD

## Abstract

Current routine histopathologic evaluation of prostate cancer does not fully account for some individual morphology patterns associated with poor outcome. Pathologists evaluate and score morphology across multiple magnifications, motivating deep learning methods to incorporate various resolutions. We have evaluated a proof-of-concept multi-view framework to classify high risk morphology architectures that does not rely on ensemble-based techniques of several single magnification models.

**Keywords:** Classification, Prostate Cancer, Digital Pathology

## 1. Introduction

**Background**   Risk stratification in localized prostate cancer is primarily determined from clinical staging and pathology grading, interpreted using the modern Gleason Grading system which characterizes histologic morphology groups (Epstein et al., 2016). Despite known grade heterogeneity, specific features such as cribriform, fused, and poorly formed glands are all considered intermediate risk group. However, there is strong evidence these morphology patterns carry differential prognostic implications (Iczkowski et al., 2011). Therefore, we aim to design a deep learning approach for enhanced risk stratification by automated classification of prostate cancer histomorphologic patterns independently associated with poor outcome.

**Related Work**   Digital pathology imaging represents a computational challenge for computer vision tasks due to large (gigapixel) images and pathologic features that are observed by pathologists at various scales/magnifications. Many deep learning algorithms operate on small patches extracted at pre-defined magnifications and derive image-level results by simple majority vote (Wei et al., 2019; Araújo et al., 2017) or additional classification ensembles (Nagpal et al., 2019; Xu et al., 2017). Recently, other works have proposed multi-magnification algorithms that either ensemble or cascade CNN models across multiple resolutions (7,8) or directly utilize features learned at intermediate CNN layers in low magnification to enrich classification at higher magnification (9).

**We propose**   a multi-view framework where each input channel represents increasing magnification views centered on patches derived from weakly-labeled annotations and train CNN classifiers using a standard ResNet architecture. We demonstrate this naive implementation, which mimics pathologist reads, matches performance of an ensemble of cascaded models from individual magnifications.

## 2. Methods

**2.1. Dataset**  56 H&E slides of radical prostatectomy specimens from 21 patients were digitized using a Zeiss AxioScan slide scanner at 20x magnification. Morphological assessment of prostate cancer architecture were mapped by a single expert genitourinary pathologist to identify 20 distinct architectural subtypes previously defined in (McKenney et al., 2016). Labels were considered weakly-annotated because regions were automatically derived from the manual annotations, which localize but do not entirely encompass morphological area (Figure 1a). For proof-of-concept, patterns were sub-grouped by prognosis into high risk (significantly worse prognosis) and other (either indeterminate or favorable prognosis) as determined by independent correlation to clinical outcome in previous study.

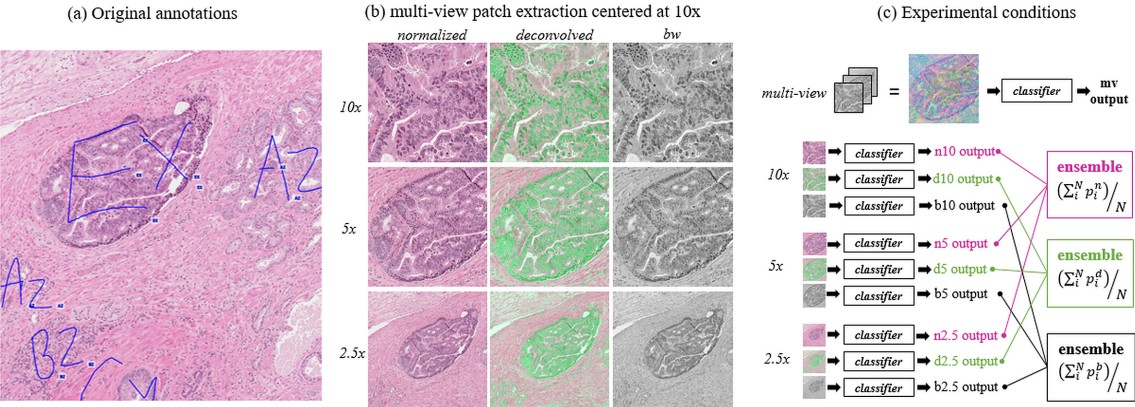

Figure 1: patch extraction. (a) weakly labeled annotations. (b) patch extraction centered on 10x magnification with colorspace modification. (c) experimental conditions for 10 individual models and multi-magnification ensemble.

**2.2. Image Processing**  300x300 pixel regions were extracted from 10x magnification with 50% overlap across the entire whole-slide image. To progressively sample morphology region, 300x300 regions centered on 10x patch coordinates were extracted at 5x and 2.5x (Figure 1b). All patch images were labeled according to annotation membership within 10x region. Three sets of colorspace images were derived at each magnification to assess influence of attention to histologic components: **1-** normalized H&E image to standardized stain matrix, **2-** deconvolved image stack consisting of hematoxylin (H), eosin (E), and black-and-white (bw) representation to enhance stromal-to-epithelial components, **3-** bw representation only. For proof-of-concept benchmark, the multi-view image was derived as bw image stack from each of the three magnifications (10x, 5x, 2.5x).

**2.3. Experiments**  Patch-based classification considered n=91612 training, n=18957 validation, and n=7255 testing images based on 15/5/1 training/validation/test split on the patient level. 10 experiment conditions are shown in Figure 1c. For each colorspace, an ensemble considering output of 2.5x, 5x, and 10x models was evaluated. Initial models were trained using ResNet-101 architecture trained using FastAI library with data augmentation (flip/rotation, contrast enhancement, mixup) and label smoothing. Individual models were

trained with weight initialization from ImageNet (75 epochs, lr=1e-6), while the multi-view was trained from scratch (200 epochs, lr=1e-7).

Table 1: Patch-Based Classification Performance

| Dataset | | H&E | | | | decon | | | | bw | | | | multi-view |
|---|---|---|---|---|---|---|---|---|---|---|---|---|---|---|
| Cohort | Metrics | 10x | 5x | 2.5x | *ensemble* | 10x | 5x | 2.5x | *ensemble* | 10x | 5x | 2.5x | *ensemble* | |
| Val | Accuracy | 0.829 | 0.825 | 0.762 | *0.870* | 0.856 | 0.814 | 0.782 | *0.871* | 0.796 | 0.782 | 0.744 | *0.859* | 0.796 |
| | AUC | **0.876** | 0.859 | 0.785 | *0.892* | 0.870 | 0.827 | 0.778 | *0.885* | 0.843 | 0.810 | 0.802 | *0.884* | 0.873 |
| Test | Accuracy | 0.865 | 0.821 | 0.917 | *0.953* | 0.903 | 0.809 | 0.856 | *0.954* | 0.861 | 0.813 | 0.826 | *0.938* | 0.893 |
| | AUC | 0.868 | 0.917 | 0.882 | *0.951* | 0.860 | 0.887 | 0.931 | *0.946* | 0.812 | 0.899 | 0.889 | *0.927* | **0.934** |

## 3. Results

Patch-based results are presented in Table 1. Among individual colorspace models, 10x models consistently outperformed 5x and 2.5x. Similarly, normalized H&E and deconvolved image stacks had higher performance than bw image classification. All ensemble models outperformed individual models. However, multi-view implementation outperformed all bw models with similar AUC to all individual models. Visual representation of whole-slide multi-view predictions for test patient are shown in Figure 2.

(a) Original image  (b) Extracted regions  (c) High Risk Ground Truth  (d) Multi-view Probability (HR)

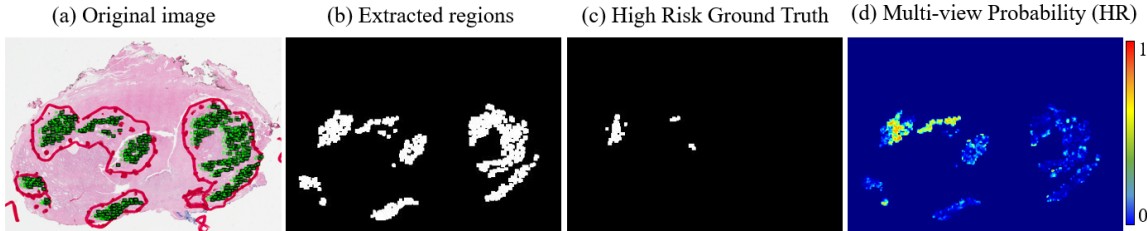

Figure 2: Whole-slide multi-view predictions of slide from unseen test patient. (a) whole slide image. (b) extracted annotation regions. (c) ground truth of high risk regions from weak labels. (d) probability distribution from sliding window demonstrating high likelihood in true positive regions.

## 4. Concluding Remarks

Our naive implementation of multi-view images utilizing bw image stacks is able to achieve similar AUC compared to any individual magnification model and decreases computational time compared to ensemble-based models. Single magnification models demonstrated deconvolved approaches highlighting epithelial-to-stroma has superior performance. This motivates further development of custom architecture for multi-view colorspace implementation.

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
