# OpenReview forum: "Multi-view Framework for Histomorphologic Classification"
_MIDL.io/2020/Conference — Submitted to MIDL 2020_

### Official Review · AnonReviewer1 · 2020-02-25
**multi view integration for histomorphologic classification**

**Rating:** 3
**Confidence:** 4

**Review:**

The quality and clarity of the work are good.
Pros:
Multiple channels of the input images are aggregated in classification.
The paper is well organized and well written.
Multi-view solution achieves better performance on testing data.
Cons:
The native channel aggregation is used in this work.
The process of "decon" is not clearly described.

---

### Official Review · AnonReviewer4 · 2020-03-03
**a multi-resolution approach to analyses histopathological slides**

**Rating:** 2
**Confidence:** 5

**Review:**

In the paper, the authors presented a multi-resolution approach to analyses histopathological slides, that mimic pathologists' work. Analysis of large scale data is challenging and new methods are necessary. The presented research goal is interesting. However, paper in the current form needs improvement to be useful for a science community.

The title of the paper is confusing. The presented approach was evaluated only on one data type (prostate cancer and evaluation was done for only one slide), by this it is difficult to estimate how good is proposed approach to other tasks. The proposed method was evaluated on a single slide, as a result, the achieved results can be overoptimistic and biased. The method description and motivation to apply selected techniques are not clear.
-	 What type of a normalization technique was applied?
-	Why color deconvolution method was applied? DL models should be able to learn correlations between colors.
-	How images were converted to BW color scale?
-	FiG.1.C is not clear. Three experiments can be distinguished on the figure, however, in the text was mentioned 10 experiments.
-	In the work, the authors applied data augmentation with label smoothing. It is not clear what does it mean. It should be explained or citations should be added.
In the result section, it is missing a comparison with a basic approach. Results for the 10x models are better for the test set only in a case of accuracy, in the term of AUC - 5x models are better.
In the paper occur formatting errors in citations.

---

### Official Review · AnonReviewer3 · 2020-03-03
**Not good enough quality for MIDL**

**Rating:** 2
**Confidence:** 5

**Review:**

In this manuscript the authors presents a multi-view  framework  to classify architectural structures in digitized pancreatic tissue samples.
Cons:
-The paper is not very well written, it's hard to follow the story.
-It's not clear what is the motivation and ultimate goal of the study.
-The experiment design is not adequate. Authors should have compared to at least one other method.

Pros:
-The idea of multi-view framework is nice and results are promising

I suggest authors to add more comparison (maybe additional data too) and a better motivation/goal statements.

---

### Official Review · AnonReviewer2 · 2020-03-13
**An evaluation of a methodology using multi-scale information for cancer risk classification in pathologic imaging**

**Rating:** 2
**Confidence:** 4

**Review:**

The authors present results using a ResNet 101 classification model to classify high risk prostate cancer from digital pathology using a mutli-scale data representation (stacking image patches at 10x, 5x, and 2.5x magnification into a single input).

Strengths:
-	The paper is well written and clinical problem is clearly defined.
-	The use of weakly annotated labels is interesting (but lacks details – see weaknesses).
-	Classification results are shown for 10 experimental conditions (10x, 5x, and 2.5x resolution each using 3 image “parameterizations”, and the multi-scale approach) along with 3 ensembling methods to demonstrate the contributions of each individual imaging component are solid.


Weaknesses:
-	While the use of weakly annotated labels is very interesting, there are no details as to how this information was utilized in the training. This has potential to be innovative, but unfortunately it is not presented here.
-	Multi-scale imaging (referred to as multi-view here) is not necessarily novel.
-	Limited testing on a single patient’s data makes it challenging to draw conclusions about how the results generalize over the entire set.


Overall, this is an interesting evaluation and application of multi-scale representation, but enthusiasm is limited due to limited testing on a single subject.

---

### Meta-Review · Area_Chair1 · 2020-04-07
**MetaReview of Paper97 by AreaChair1**

**Rating:** 1

**Metareview:**

The majority of reviewers recommended reject based on lack of details and proper comparison. There was no rebuttal submitted.

**Paper Type:**

validation/application paper

---

### Decision · Program_Chairs · 2020-04-11

Reject